# Bone marrow involvement in patients with metastatic castration sensitive prostate cancer

Mohammed Shahait[1], Ramiz Abu-hijlih[2], Alaa Salamat[3], Nassib Abou Heidar[4], Baha' Sharaf[3], Fawzi Abuhijla[2], Samer Salah[3]*

1 Department of Surgery, King Hussein Cancer Center, Amman, Jordan, 2 Department of Radiation Oncology, King Hussein Cancer Center, Amman, Jordan, 3 Department of Medical Oncology, King Hussein Cancer Center, Amman, Jordan, 4 Department of Surgery, American University of Beirut Medical Center, Beirut, Lebanon

* samer.salahmd@gmail.com

**Data Availability Statement:** All relevant data are within the paper and its Supporting Information files.

## Abstract

### Introduction

The clinical significance of bone marrow (BM) metastasis in prostate cancer as well as impact on oncological prognosis is unclear. We aim to assess the prevalence and clinical outcomes of BM metastasis at initial presentation of metastatic castrate sensitive prostate cancer (CSPC).

### Patients and methods

Retrospective chart review of newly diagnosed metastatic CSPC patients was performed with collection of clinicopathologic and radiologic characteristics. Descriptive univariate and multivariate analysis was performed as well as survival measures (OS and PFS), which was done using the Kaplan-Meier survival and the Log-rank test.

### Results

189 patients were eligible, of which, eleven patients (6%) had biopsy proven BM involvement at diagnosis. There was a trend to poorer PFS and OS in patients with BM involvement but not statistically significant; however, factors that correlated with inferior PFS and OS in the multivariate analysis included ECOG PS, ALP, and Hb.

### Conclusion

BM metastasis in prostate cancer may lead to poorer survival. Clinical features including poor performance status, anemia, and elevated ALP, could guide bone marrow biopsies in the future to diagnose bone marrow metastasis at an earlier stage.

**Funding:** The authors received no specific funding for this work.

**Competing interests:** All the authors declare no conflict of interest.

## Introduction

Prostate cancer is among the most common cancers in men worldwide and stage at initial diagnosis is the most important determinant of survival [1, 2]. Non-metastatic disease is associated with 10-year overall survival (OS) rate that exceeds 90% [2]. In fact, patients with low risk groups can have a normal life-span and may be managed with active surveillance [3]. Conversely, patients with metastatic disease have poorer prognosis. The survival of patients with metastatic disease is variable as many factors play role in oncologic outcomes. Prognostic factors that correlate with poorer oncologic outcomes for patients with metastatic disease include visceral metastasis, poor performance status, high serum alkaline phosphatase, pain at presentation, and the burden of metastatic disease [4]. It remains unclear whether bone marrow (BM) metastasis at initial presentation would similarly have a negative impact on outcomes.

Bone marrow provides a fertile microenvironment for metastatic tumor cells. Some studies suggest that bone metastatic prostate cancer cells parasitize the microenvironment and survive within BM [5]. As a result of the low frequency of BM metastasis at initial presentation of metastatic prostate cancer, BM biopsy is not routinely performed, and is reserved for patients with unexplained cytopenias [4, 6, 7].

Detection of BM metastasis heralds poor prognosis in variety of solid malignancies including prostate cancer [8, 9]. However, metastasis to BM typically occurs late in the course of metastatic castration resistant disease. Data addressing the clinical significance of BM metastasis at initial diagnosis of metastatic castration sensitive prostate cancer (CSPC) is scarce. Furthermore, whether BM metastasis predicts early transition to castration resistance and short survival is still unclear. Moreover, whether BM metastasis at initial presentation should be managed with early therapy intensification similar to high risk groups of patients with visceral metastasis and high volume disease, is currently unknown [10]. Owing to such limited data, we sought to assess the prevalence and clinical outcomes of BM metastasis at initial presentation of metastatic castrate sensitive prostate cancer.

## Materials and methods

### Objectives

This study aims at describing the frequency of BM metastasis in newly diagnosed patients with metastatic CSPC. Furthermore, we sought to assess the impact of BM metastasis on time to PSA progression and Overall survival (OS).

### Patients

Data of newly diagnosed patients with metastatic CSPC presenting to our center from January, 2010 to December, 2019, were retrospectively collected. To be eligible, patients were required to have metastatic disease detected on conventional imaging studies including computed tomography (CT) scan, Magnetic resonance imaging (MRI) or bone scan. Patients were also required to have castration sensitive disease. As such, patients who were referred to us with castration resistant metastatic prostate cancer were excluded. Similarly, patients who developed metastatic disease following non-metastatic (M0) castration resistant disease were excluded.

Data of eligible patients were retrospectively abstracted from the electronic medical records following acquisition of an institutional review board approval. Data of interest included: age, date of diagnosis of the metastatic disease, any prior local therapy to prostate, Gleason Score, sites of metastasis, number and distribution of metastatic bone lesions, eastern cooperative oncology group performance status (ECOG PS), details of initial systemic therapy (ADT alone

or ADT plus docetaxel or abiraterone). In addition, we referred to the baseline laboratory records to collect data on blood counts, results of any BM aspirate and biopsy, serum PSA, and serum alkaline phosphatase (ALP) value. Finally, we gathered data on time of first PSA progression after starting ADT, and dates of last follow up or death. PSA progression was defined according to the prostate cancer working group criteria (PCWGC 3) as $\geq$ 25% increase of serum PSA from the nadir value following commencement of ADT, that has to be confirmed with a subsequent value done at least 3 weeks later [11].

Metastatic burden was stratified into high volume and low volume disease based on the definition from CHAARTED trial. Accordingly, high volume disease was defined as the presence of more than three metastatic bone lesions with at least one lesion outside the spine and pelvis or the presence of visceral metastasis, while the LATITUDE trial defined risk status was used to classify patients into high-risk/low-risk disease, where high-risk disease is defined as having at least two of the following: GS $\geq$8, at least three bone lesions, visceral metastasis [12, 13].

Diagnosis of BM metastasis was made by pathologic evaluation of BM aspirate and trephine biopsy specimens, revealing infiltration of BM with prostatic adenocarcinoma cells.

### Statistical analysis

Descriptive statistics were utilized when appropriate to report means, median, standard deviations, and proportions. The independent sample *t*-test and the chi-square test were used for comparison of means and proportions, respectively. Overall survival was calculated from the date of diagnosis of metastatic disease until last follow up or death. PSA progression free survival (PFS) was calculated from the time of initiation of ADT until the first documentation of PSA progression according to PCWGC 3. PFS and OS were estimated by the Kaplan-Meier survival method. Survival comparisons were carried out by the Log-rank test. Further, we assessed the effect of the following factors on PSA PFS and OS: presence of BM involvement, age, Gleason score, number of organ metastasis, whether the metastasis is visceral, serum PSA, ECOG PS, serum ALP, serum Hb, whether the patients had received prior local therapy to prostate, and whether upfront therapy added to ADT was administered (docetaxel or abiraterone). All significant factors with effect on OS ($p < 0.05$) in the univariate analysis were entered into a multivariate analysis to assess for independent factors that could predict OS utilizing the cox-regression method. All statistical analyses were performed using SPSS version 19 (SPSS Inc., Chicago, IL).

### Results

#### Patients

We initially identified a total of 280 newly diagnosed metastatic prostate cancer patients treated at our center during the eligibility period. Forty-three (15%) were excluded because they presented with mCRPC, 17 (6%) because they were referred to us to get a second opinion then they lost follow up, and 9 (3%) due to lack of pathologic confirmation of prostatic adenocarcinoma. In addition, 4 (1%) were excluded for the diagnosis of a second cancer, 4 (1%) because the metastatic disease was not detected by conventional imaging studies, and 14 (5%) for missing follow up data. Thus, a total of 189 patients remained eligible for analysis. Of the 189 patients, eleven patients (6%) had biopsy proven bone marrow involvement on initial diagnosis of the metastatic disease. The clinical characteristics of the patients at baseline are summarized in Table 1.

The median follow-up time of the cohort was 42.40 months. Patients with BM metastasis were more likely to have $\geq$ 2 organ metastasis, anemia, and high serum alkaline phosphatase (ALP) (Table 1). In addition, patients with BM involvement had worse ECOG PS compared to

**Table 1. Characteristics of patients with metastatic castrate sensitive prostate cancer with bone marrow involvement and without bone marrow involvement.**

| name | value | Total | Bone marrow | | P-value |
|---|---|---|---|---|---|
| | | | no (*n* = 178) | yes (*n* = 11) | |
| ALP cutoff 130 | N/A | 1(0.5%) | 1(0.6%) | | 0.027 |
| | < 130 | 99(52.7%) | 97(54.8%) | 2(18.2%) | |
| | ≥130 | 89(47.3%) | 80(45.2%) | 9(81.8%) | |
| Number of organ metastasis | >1 | 85(45.0%) | 76(42.7%) | 9(81.8%) | 0.014 |
| | one | 104(55.0%) | 102(57.3%) | 2(18.2%) | |
| Visceral metastasis | no | 156(82.5%) | 148(83.1%) | 8(72.7%) | 0.410 |
| | yes | 33(17.5%) | 30(16.9%) | 3(27.3%) | |
| ECOG cutoff | N/A | 23(12.2%) | 23(12.9%) | | 0.057 |
| | ≥ 1 | 84(50.6%) | 75(48.4%) | 9(81.8%) | |
| | zero | 82(49.4%) | 80(51.6%) | 2(18.2%) | |
| Calcium cutoff | N/A | 5(2.6%) | 5(2.8%) | | 1.000 |
| | < 10.4 | 180(97.8%) | 169(97.7%) | 11 (100%) | |
| | ≥ 10.4 | 4 (2.2%) | 4 (2.3%) | | |
| Hb cutoff | < 12 | 50(26.5%) | 43(24.2%) | 7(63.6%) | 0.009 |
| | ≥12 | 139(73.5%) | 135(75.8%) | 4(36.4%) | |
| Platelets | low platelets | 17 (9.0%) | 15 (8.4%) | 2(18.2%) | 0.259 |
| | normal platelets | 172(91.0%) | 163(91.6%) | 9(81.8%) | |
| Prior local therapy | N/A | 1(0.5%) | 1(0.6%) | | 0.389 |
| | no | 180(95.7%) | 170(96.0%) | 10(90.9%) | |
| | yes | 8 (4.3%) | 7 (4.0%) | 1 (9.1%) | |
| Number of bone lesions | N/A | 17(9.0%) | 16(9.0%) | 1(9.1%) | 0.172 |
| | < 3 | 37(21.4%) | 37(22.7%) | 0(0%) | |
| | ≥3 | 135(78.0%) | 125(76.7%) | 10(90.9%) | |
| High volume | N/A | 1 (0.5%) | 1 (0.6%) | | 0.225 |
| | no | 41(21.7%) | 41(23.0%) | 0(0%) | |
| | not assessable (no baseline bone scan) | 1 (0.5%) | 1 (0.6%) | | |
| | yes | 146(77.2%) | 135(75.8%) | 11 (100%) | |
| High risk | no | 44(23.3%) | 44(24.7%) | 0(0%) | 0.124 |
| | not assessable | 1 (0.5%) | 1 (0.6%) | | |
| | yes | 144(76.2%) | 133(74.7%) | 11 (100%) | |
| Upfront docetaxel or abiraterone | no | 134(70.9%) | 127(71.3%) | 7(63.6%) | 0.733 |
| | yes | 55(29.1%) | 51(28.7%) | 4(36.4%) | |

patients without BM involvement; however, p-value did not reach statistical significance (Table 1).

Regarding metastatic burden 146 (77.2%) patients had high volume disease, and 41 (23%) had low volume disease; 2 patients were not assessable for volume status. A total of 144 patients (76.2%) had LATITUDE-defined high-risk disease, 44 (23.3%) had low risk disease, and one was not assessable. All patients with BM involvement had high volume and high risk disease.

## Effect of BM metastasis on PFS

Median PSA PFS for the entire cohort was 13.8 months (95% CI: 11.1–16.5 months). There was no statistically significant difference in median PSA PFS between patients with and without BM involvement; 9.4 and 17.1 months respectively, *p* = 0.29 (Fig 1). Factors that correlated with inferior PSA PFS were low performance status ECOG ≥1, presence of ≥ 3 bone lesions, high ALP and Hb < 12 g/l (Table 2).

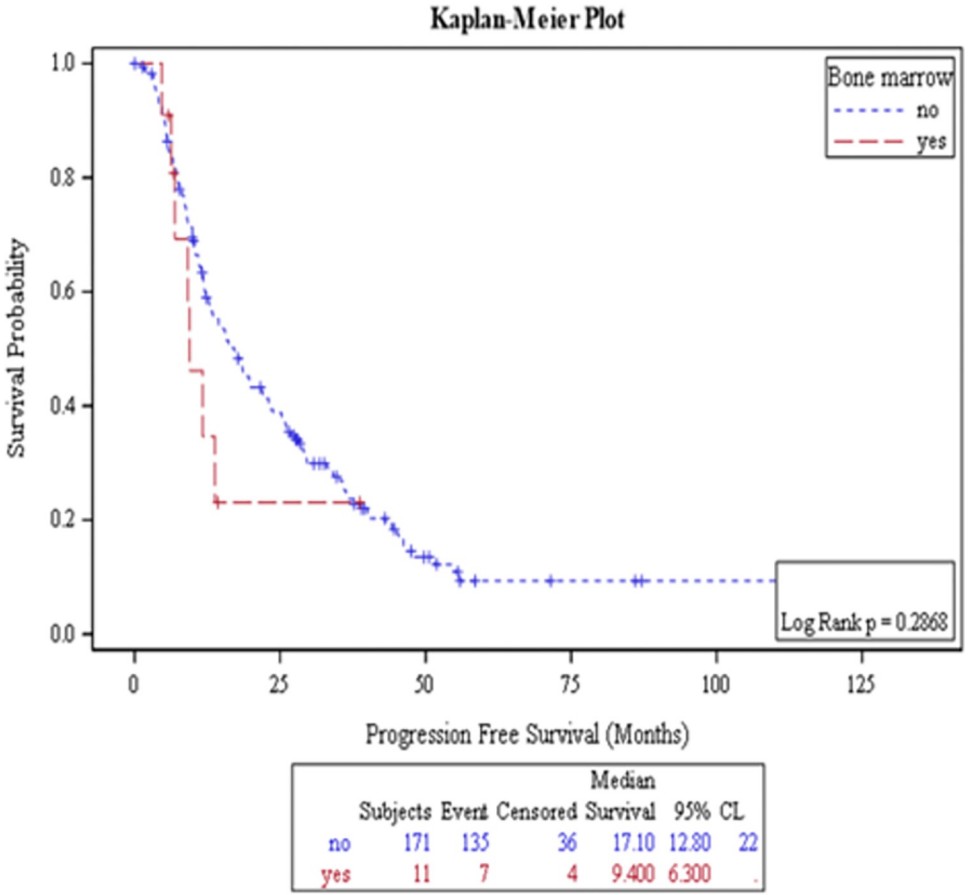

**Fig 1. Kaplan-Meier estimation of time to PSA progression according to presence of bone marrow metastasis.**

We did not find a statistically significant difference in median PSA PFS for patients with BM metastasis who received upfront docetaxel or abiraterone compared to patients with BM metastasis who received ADT alone, 11.7 and 6.3 months respectively, $p = 0.63$.

**Table 2. Association between bone marrow involvement and PSA progression free survival on multivariate analysis.**

| Parameter | | | Hazard Ratio | 95% Hazard Ratio Confidence Limits | |
|---|---|---|---|---|---|
| Gleason Score | $\geq 8$ vs. $< 8$ | | 1.194 | 0.755 | 1.891 |
| Number of organ metastasis | $\geq$ one vs. One | | 1.224 | 0.879 | 1.702 |
| Number of bone lesion | $\geq 3$ vs. $< 3$ | | 2.633 | 1.629 | 4.257 |
| Bone marrow | Yes vs. No | | 1.516 | 0.704 | 3.264 |
| Visceral metastasis | Yes vs. No | | 0.880 | 0.558 | 1.386 |
| PSA | PSA$> = 154.5$ vs. PSA$<154.5$ | | 1.215 | 0.869 | 1.699 |
| ECOG cutoff | zero vs. $\geq 1$ | | 0.291 | 0.198 | 0.427 |
| ALP cutoff | $\geq$130 vs. $< 130$ | | 1.881 | 1.348 | 2.625 |
| Hb cutoff | $\geq$12 vs. less than 12 | | 0.474 | 0.328 | 0.686 |
| Prior local therapy | Yes vs. No | | 1.711 | 0.748 | 3.914 |
| Upfront docetaxel | Yes vs. No | | 0.894 | 0.623 | 1.283 |
| Age | Age$> = 69$ vs. Age$<69$ | | 0.891 | 0.641 | 1.240 |
| Platelets | normal platelets vs. low plat | | 1.453 | 0.763 | 2.766 |

### Effect of BM metastasis on OS

Median overall survival (OS) was 42.2 months for the entire cohort (95% CI: 33.0–51.2 months). Factors that significantly predicted worse OS in the multivariate analysis included ECOG PS, ALP, and Hb. (Table 3). We observed a non-significant trend for inferior OS for patients with BM metastasis compared to patients without BM metastasis; median OS was 18.1 and 42.1 months respectively, $p = 0.055$.

## Discussion

Prostate cancer remains as one of the most frequently diagnosed malignancies in men and a leading cause of death worldwide [2]. However, to add to the complexity of management of men with prostate cancer, this disease has a vast clinical variability and ranges from very indolent disease to rapidly progressive metastasis and castration resistance. There have been many efforts in the last decades to stratify patients using clinical parameters as well as biomarkers to stratify patients in all clinical stages to predict prognosis in order to guide management strategies [12].

In the case of metastatic prostate cancer, some nomograms have been developed to accurately predict prognosis of the disease. For instance, Hou et al. proposed a predictive nomogram using age, PSA score, Gleason grade, stage, as well as other clinical variables in patients with mainly bone metastasis [13]. Jiang et al. also devised a similar nomogram with relevant clinical and pathologic features to predict survival [14]. However, we still to this day encounter patients who progress in metastasis more rapidly than patients with similar clinical features. The authors have proposed that bone marrow metastasis in patients with prostate cancer could be a variable that might explain this rapid progression.

Bone marrow metastasis have been shown to be an adverse feature of many visceral malignancies [15, 16]. In most series published on bone marrow metastasis survival does not exceed weeks [16–18]. Therefore, the authors propose early identification of prostate cancer patients with bone marrow metastasis to stratify patients with more aggressive disease to guide therapy. Having said that, there are no criteria to guide decision making for performing bone marrow biopsy or aspirate and to guide subsequent management.

In our series, we found that patients with bone marrow metastasis had a worse ECOG performance status as well as a lower hemoglobin level and a higher ALP level. These patients conferred a significantly worse prognosis in terms of PSA-progression free survival as well as overall survival. The significance in this study is that the authors have shown that bone marrow metastasis confers worse oncologic outcomes on one hand, and the criteria for suspicion of bone marrow metastasis which include anemia, worse performance status, and elevated ALP.

Although our study is the first to explore the prognostic role of bone marrow involvement in patients with mCSPC, there are many limitations that we acknowledge. First, the small sample size and the retrospective design of the study are recognized limitations. Second, the study includes patients treated over 12 years, in which disease presentation and management have evolved. Finally, all of these patients were referred to our institution with bone morrow biopsy/aspirate was done in other institutions; therefore, we were not able to control for selection bias and other factors affected the decision of the referring physician to perform the bone marrow biopsy.

In conclusion, bone marrow metastasis in prostate cancer seems to lead to significantly poorer survival. Further studies are, however, needed to further confirm our findings. Moreover, clinical features, found to be significant among patients with bone marrow metastasis (including poor performance status, anemia, and elevated ALP) could guide bone marrow biopsies in the future to diagnose bone marrow metastasis at an earlier stage.

**Table 3. Association between bone marrow involvement and overall survival on multivariate analysis.**

| Parameter | | | Hazard Ratio | 95% Hazard Ratio Confidence Limits | |
|---|---|---|---|---|---|
| Gleason Score | ≥8 vs. < 8 | | 1.906 | 1.049 | 3.465 |
| Number of organ metastasis | ≥one vs. One | | 1.299 | 0.881 | 1.915 |
| Number of bone lesion | ≥ 3 vs. < 3 | | 2.472 | 1.408 | 4.340 |
| Bone marrow | Yes vs. No | | 1.411 | 0.616 | 3.231 |
| Visceral metastasis | Yes vs. No | | 1.108 | 0.658 | 1.868 |
| PSA | PSA> = 154.5 vs. PSA<154.5 | | 1.370 | 0.925 | 2.031 |
| ECOG cutoff | zero vs. ≥ 1 | | 0.159 | 0.098 | 0.258 |
| ALP cutoff | ≥130 vs. < 130 | | 1.973 | 1.337 | 2.911 |
| Hb cutoff | ≥12 vs. < 12 | | 0.316 | 0.210 | 0.476 |
| Prior local therapy | Yes vs. No | | 0.978 | 0.359 | 2.666 |
| Upfront docetaxel | Yes vs. No | | 0.730 | 0.454 | 1.174 |
| Age | Age> = 69 vs. Age<69 | | 1.087 | 0.739 | 1.599 |
| platelets | normal platelets vs. low plat | | 1.193 | 0.579 | 2.461 |

## Supporting information

**S1 Data.**
(XLSX)

## Author Contributions

**Conceptualization:** Mohammed Shahait, Alaa Salamat, Nassib Abou Heidar, Samer Salah.

**Data curation:** Mohammed Shahait, Ramiz Abu-hijlih, Baha' Sharaf.

**Formal analysis:** Alaa Salamat.

**Methodology:** Nassib Abou Heidar, Baha' Sharaf, Samer Salah.

**Resources:** Baha' Sharaf.

**Supervision:** Ramiz Abu-hijlih, Fawzi Abuhijla.

**Validation:** Mohammed Shahait, Fawzi Abuhijla.

**Visualization:** Nassib Abou Heidar, Samer Salah.

**Writing – original draft:** Mohammed Shahait, Ramiz Abu-hijlih, Nassib Abou Heidar.

**Writing – review & editing:** Mohammed Shahait, Alaa Salamat, Nassib Abou Heidar, Baha' Sharaf, Fawzi Abuhijla, Samer Salah.

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
