## [Decision Letter · Decision Letter 0]

4 Mar 2022

PONE-D-21-40910“Bone Marrow Involvement in Patients with Metastatic Castration Sensitive Prostate Cancer”PLOS ONE

Dear Dr. Salah,

Thank you for submitting your manuscript to PLOS ONE. After careful consideration, we feel that it has merit but does not fully meet PLOS ONE’s publication criteria as it currently stands. Therefore, we invite you to submit a revised version of the manuscript that addresses the points raised during the review process.

We look forward to receiving your revised manuscript.

Kind regards,

Matteo Bauckneht, MD, PhD

Academic Editor

PLOS ONE

Journal Requirements:

“No Funding”

Reviewers' comments:

Reviewer's Responses to Questions

**Comments to the Author**

1. Is the manuscript technically sound, and do the data support the conclusions?

Reviewer #1: Yes

Reviewer #2: Yes

2. Has the statistical analysis been performed appropriately and rigorously? 

Reviewer #1: Yes

Reviewer #2: Yes

3. Have the authors made all data underlying the findings in their manuscript fully available?

Reviewer #1: Yes

Reviewer #2: Yes

4. Is the manuscript presented in an intelligible fashion and written in standard English?

Reviewer #1: Yes

Reviewer #2: Yes

5. Review Comments to the Author

Reviewer #1: Very interesting idea, specially in the setting of metastatic castrate sensitive prostate cancer. Different approaches are being developed to subselect patients at higher risk of progression before progressing to crpc state.

It was not clear in the abstract and the full manuscript if all the 189 patients subject of the study, underwent a bone marrow biopsy. I am assuming it is the case. But I think it should be mentioned.

Reviewer #2: The article is a retrospective review of clinical data that aims to determine the clinical importance of bone marrow metastasis of prostate cancer on the patient prognosis and survival. It is an important topic, the analysis includes a relevant number of patients and is based on observation of images clinical and laboratory data.

The article is well written, with a good flow and easy to read for a non-expert audience, but also with sufficient technical depth.

The statistical analysis is appropriate, and the conclusions correspond to the clinical observations. The authors acknowledge their limitations accordingly, with scientific soundness.

The novelty of the article is moderate but adds relevant information to the field that sums to the overall knowledge of prostate cancer.

The absence of typos and grammar issues also reveals that the authors performed an appropriate and extensive proof reading,

The article deserves to be published and this reviewer have one minor edit in the article:

The article would increase its relevance including images of the bone metastasis of prostate cancer that were used in the diagnosis MRI or CT scans.

God luck in your following research

6. PLOS authors have the option to publish the peer review history of their article (what does this mean?). If published, this will include your full peer review and any attached files.

Reviewer #1: **Yes: **Ali Merhe

Reviewer #2: No

---

## [Author Response · Author response to Decision Letter 0]

18 May 2022

Dr. Bauckneht

Managing editor

PLOS ONE

RE: “Bone Marrow Involvement in Patients with Metastatic Castration Sensitive Prostate Cancer”

 (Manuscript ID PONE-D-21-40910)

Dear Dr. Bauckneht:

Thank you for considering our manuscript for possible publication. We appreciate the comments of the reviewers and we have revised the manuscript accordingly. 

Hereby the revised version of the : “Bone Marrow Involvement in Patients with Metastatic Castration Sensitive Prostate Cancer” (Manuscript ID PONE-D-21-40910)

We have listed the concerns of the reviewers and have detailed our actions to address these issues, point by point. 

To summarize the most important changes, we have:

-We updated the methodology section to include the ethical approval. 

-We added a statement regarding the data availability.

Finally, we want to thank you and the reviewers, for the effort to review this manuscript. 

I hope we were able to address all reviewers’ concerns satisfactorily. 

Looking forward to hearing back soon from you. 

Sincerely, 

Samer Salah

Response:

“No Funding”

Response:

None

Response:

None

Response:

None

Response:

The statement was added “The authors received no specific funding for this work.”

Response:

We totally believe in the importance of data availability. However, the IRB committee at our institution did not grant us the right to publicly sharing the data as there are no legislation in our country to regulate sharing such data publicly; and this might put the authors and the institution at high legal liability in our country. However, the data might be available if requested , by contacting the IRB office at King Hussein Cancer Center to grant us approval of data sharing. Contacting the IRB will commence once you request the data.

Response:

Added Samer Salah : 0000-0001-7421-2522

Response:

Moved to the method section

Response:

The references were double checked and complies with the journal style.

Reviewers' comments:

Reviewer's Responses to Questions

Comments to the Author

1. Is the manuscript technically sound, and do the data support the conclusions?

Reviewer #1: Yes

Reviewer #2: Yes

2. Has the statistical analysis been performed appropriately and rigorously?

Reviewer #1: Yes

Reviewer #2: Yes

3. Have the authors made all data underlying the findings in their manuscript fully available?

Reviewer #1: Yes

Reviewer #2: Yes

4. Is the manuscript presented in an intelligible fashion and written in standard English?

Reviewer #1: Yes

Reviewer #2: Yes

5. Review Comments to the Author

Reviewer #1: Very interesting idea, specially in the setting of metastatic castrate sensitive prostate cancer. Different approaches are being developed to subselect patients at higher risk of progression before progressing to crpc state.

It was not clear in the abstract and the full manuscript if all the 189 patients subject of the study, underwent a bone marrow biopsy. I am assuming it is the case. But I think it should be mentioned.

Response:

Thank you for your comment. Actually, BM bx is not a standard of care in our institution and we don’t perform BM on all these patients. It was a subset of patients who had BM bx in other institution for unknown reasons and referred for treatment at ours.

We added this statement to the methodology section :” Only eleven patients (6%) had bone marrow biopsy in other institutions and were referred to our institution for prostate cancer management.”

Also we highlighted this in our limitation section :Finally, all of these patients were referred to our institution with bone morrow biopsy/aspirate was done in other institutions; therefore, we were not able to control for selection bias and other factors affected the decision of the referring physician to perform the bone marrow biopsy.

Reviewer #2: The article is a retrospective review of clinical data that aims to determine the clinical importance of bone marrow metastasis of prostate cancer on the patient prognosis and survival. It is an important topic, the analysis includes a relevant number of patients and is based on observation of images clinical and laboratory data.

The article is well written, with a good flow and easy to read for a non-expert audience, but also with sufficient technical depth.

The statistical analysis is appropriate, and the conclusions correspond to the clinical observations. The authors acknowledge their limitations accordingly, with scientific soundness.

The novelty of the article is moderate but adds relevant information to the field that sums to the overall knowledge of prostate cancer.

The absence of typos and grammar issues also reveals that the authors performed an appropriate and extensive proof reading,

The article deserves to be published and this reviewer have one minor edit in the article:

The article would increase its relevance including images of the bone metastasis of prostate cancer that were used in the diagnosis MRI or CT scans.

Response:

Thank you for your comments. BM metastasis was based on BM biopsy aspirate not images.

---

## [Editor Report · Decision Letter 1]

22 Jun 2022

“Bone Marrow Involvement in Patients with Metastatic Castration Sensitive Prostate Cancer”

PONE-D-21-40910R1

Dear Dr. Salah,

We’re pleased to inform you that your manuscript has been judged scientifically suitable for publication and will be formally accepted for publication once it meets all outstanding technical requirements.

Kind regards,

Matteo Bauckneht

Academic Editor

PLOS ONE
---

## [Editor Report · Acceptance letter]

27 Jun 2022

PONE-D-21-40910R1 

Bone Marrow Involvement in Patients with Metastatic Castration Sensitive Prostate Cancer 

Dear Dr. Salah:

I'm pleased to inform you that your manuscript has been deemed suitable for publication in PLOS ONE. Congratulations! Your manuscript is now with our production department. 

Kind regards, 

on behalf of

Dr. Matteo Bauckneht 

Academic Editor

PLOS ONE